# Supplementation with a Salmon Bone Complex (CalGo^®^) Preserves Femoral Neck BMD and Attenuates Lumbar Spine Loss: A 24-Month Randomized, Placebo-Controlled Trial

**DOI:** 10.3390/biomedicines13112616

**Published:** 2025-10-25

**Authors:** Christian Bjerknes, Anne Rørvik Standal, Crawford Currie, Bomi Framroze, Tor Åge Myklebust, Tommy Frøseth Aae, Erland Hermansen

**Affiliations:** 1Hofseth Biocare ASA, Keiser Wilhelms Gate 24, 6003 Ålesund, Norway; anst@hofsethbiocare.no (A.R.S.); cc@hofsethbiocare.no (C.C.); bf@hofsethbiocare.no (B.F.); 2Department of Registration, Cancer Registry of Norway, Norwegian Institute of Public Health, 0379 Oslo, Norway; tor.age.myklebust@fhi.no; 3Department of Research and Innovation, Møre og Romsdal Hospital Trust, 6026 Ålesund, Norway; tommy.aae@gmail.com (T.F.A.); erland.hermansen45@gmail.com (E.H.); 4Department of Neuromedicine and Movement Science, Faculty of Medicine and Health Sciences, Norwegian University of Science and Technology (NTNU), 7491 Trondheim, Norway; 5Department of Health Sciences, Faculty of Medicine and Health Sciences, Norwegian University of Science and Technology (NTNU), P.O. Box 1600, 6026 Ålesund, Norway

**Keywords:** salmon bone meal, bone mineral density, BMD, DXA, menopause, osteopenia, osteoporosis

## Abstract

**Background/Objectives**: Osteopenia is common in postmenopausal women and predisposes to osteoporosis and fracture, representing a population at risk of bone loss but without indication for pharmacologic therapy. Conventional calcium salts offer modest, often transient gains in bone mineral density (BMD). We evaluated whether CalGo^®^, a salmon bone complex containing microcrystalline hydroxyapatite within a collagen-rich matrix, preserves BMD versus placebo in post-menopausal women with osteopenia. **Methods**: In a 24-month, randomized, double-blind, placebo-controlled trial, 80 women (50–80 years) with dual-energy X-ray absorptiometry (DXA)-confirmed femoral-neck osteopenia were assigned to CalGo^®^ (2 g/day) or placebo. The prespecified primary endpoint was 24-month change in femoral-neck BMD (g/cm^2^) analyzed by linear regression (unadjusted and baseline-adjusted). Secondary endpoints included lumbar spine and distal radius BMD, serum P1NP and β-CTX-I, health-related quality of life, and safety. **Results**: The primary analysis included participants with 24-month DXA (CalGo^®^ *n* = 29; placebo *n* = 30). Femoral-neck BMD was maintained with CalGo^®^ (+0.003 g/cm^2^; +0.4%) but declined with placebo (−0.017 g/cm^2^; −2.4%), yielding a significant baseline-adjusted between-group difference of +0.019 g/cm^2^ (95% confidence interval (CI) 0.001–0.038; *p* = 0.044). Lumbar-spine loss was attenuated with CalGo^®^ (−0.005 g/cm^2^; −0.3%) versus placebo (−0.028 g/cm^2^; −3.4%); the adjusted difference favored CalGo^®^ (+0.026 g/cm^2^; *p* = 0.058). In exploratory responder analysis, ≥1% lumbar-spine gain was more likely with CalGo^®^ (32.5% vs. 11.4%; OR 3.61; *p* = 0.043). No treatment effects were observed at the distal radius, in P1NP or β-CTX-I, or in EQ-5D-3L/EQ-VAS. CalGo^®^ was well tolerated with no hepatic or renal safety signals. **Conclusions**: CalGo^®^ maintained femoral-neck bone mineral density and reduced lumbar-spine loss over 24 months in osteopenic women, with good tolerability. These findings support its potential role as a nutritional approach for maintaining bone health.

## 1. Introduction

Bone mass typically begins to decline from the third decade of life, with the rate of loss accelerating during the menopausal transition, increasing the risk of osteopenia [1]. Osteopenia often progresses to osteoporosis, characterized by reduced bone strength and a higher risk of fractures at the femoral neck, spine, and distal radius [2,3]. As bone loss advances and fracture risk rises, preventive strategies have focused on optimizing calcium and vitamin D intake. Calcium supplementation—typically co-administered with vitamin D—remains a cornerstone of bone health, though meta-analyses indicate that its effects are modest and plateau over time [4]. Pooled data show site-specific gains of approximately 0.7–1.8% in spinal and hip bone mineral density (BMD) over 1–2 years, with little further increase thereafter and an overall fracture risk reduction of about 12% (risk ratio [RR] ≈ 0.88) compared with placebo. Greater benefits are observed when intake exceeds 1200 mg/day calcium and 800 IU/day vitamin D with good adherence, although gastrointestinal intolerance—most commonly constipation and bloating—often limits higher dosing [5,6,7].

Despite these modest benefits, variability across trials remains substantial, and calcium supplementation may transiently elevate serum calcium concentrations, which Mendelian randomization studies have associated with an increased risk of cardiovascular disease [4]. This concern has renewed interest in food-based calcium sources that provide a more gradual systemic absorption. In a recent randomized controlled trial, meal replacements fortified with eggshell calcium and vitamin D attenuated femoral-neck bone loss over 6–12 months in postmenopausal women, reinforcing the translational potential of gently absorbed, naturally derived calcium sources to sustain bone density [8]. Similarly, microcrystalline hydroxyapatite (MCH) has been shown to induce smaller post-dose serum calcium increases than conventional calcium salts while maintaining comparable effects on bone turnover [9]. These findings suggest that physiologic, matrix-bound calcium sources—which more closely emulate the mineral–organic structure of bone—may offer improved skeletal support, particularly when co-delivered with collagen.

The addition of collagen, another major bone component, appears to enhance the effects of natural calcium supplementation. A systematic review and meta-analysis of randomized controlled trials demonstrated that ossein–hydroxyapatite complex (OHC), a bovine-derived bone matrix preparation, was significantly more effective than calcium carbonate in maintaining trabecular bone mineral density, with a pooled weighted mean difference of approximately +1% in favor of OHC [10]. Subsequent trials confirmed greater preservation of lumbar spine BMD in postmenopausal women treated with OHC over up to three years, although study heterogeneity limits the generalizability of the results [11]. More recently, a large prospective study in perimenopausal women found lumbar spine BMD to remain stable with OHC but to decline by 3.1% with calcium carbonate, with fewer adverse reactions in the OHC group [12]. Collectively, these data support the concept that calcium delivered within a collagen-rich matrix enhances both efficacy and tolerability, providing a biological rationale for evaluating analogous preparations derived from alternative natural sources.

CalGo^®^ is a marine-derived bone matrix biocomposite in which microcrystalline hydroxyapatite is physically integrated with collagen and native trace minerals. Conceptually analogous to bovine OHC, CalGo^®^ differs by being sourced from Norwegian Atlantic salmon (*Salmo salar*) and produced without harsh acid or alkali demineralization, thereby preserving the native mineral–organic architecture of bone. This process yields a nutrient profile that closely resembles the composition of natural bone, offering potential benefits for skeletal maintenance.

The functional potential of this matrix is supported by preclinical and early clinical studies. In vitro, salmon bone powder (SBP) stimulated osteoblast proliferation more strongly than calcium from algae, eggshell, or synthetic salts, with osteoinductive activity also demonstrated in an in vivo osteoporosis model [13,14]. Early clinical data indicated approximately sixfold higher absorbability of SBP-derived calcium hydroxyapatite compared with calcium carbonate [15]. Importantly, serum calcium remained within the normal range and calcifediol levels were unchanged, suggesting improved bioavailability without increased renal excretion.

Building on this compositional and preclinical rationale, we targeted women with osteopenia (−2.5 < T-score ≤ −1.0), a population at measurable risk of progressive bone loss but without indication for pharmacologic therapy [16,17]. Osteopenic women, particularly during the peri- to postmenopausal transition, experience accelerated bone loss due to estrogen decline, providing a biologically and clinically relevant setting for evaluating preventive nutritional strategies. We therefore conducted a 24-month, randomized, double-blind, placebo-controlled trial to evaluate the efficacy and safety of CalGo^®^ in preserving bone mineral density in post-menopausal women. The primary endpoint was change in femoral-neck BMD, with secondary endpoints including lumbar spine and distal radius BMD, bone-turnover markers, health-related quality of life, and safety. We hypothesized that CalGo^®^ supplementation would maintain femoral-neck BMD compared with placebo.

## 2. Materials and Methods

### 2.1. Trial Design and Registration

This investigation was conducted as a randomized, double-blind, placebo-controlled, multicenter clinical trial at three Norwegian sites: Ålesund/Medi3, Kristiansund Hospital, and Lovisenberg Diakonale Sykehus (LDS). This study was carried out in collaboration between Hofseth BioCare ASA (Ålesund, Norway) and the Møre and Romsdal Health Trust, and is registered at ClinicalTrials.gov (Identifier: NCT05066477). Eligible participants were recruited through mailed invitations, posters, and public announcements, and were randomly assigned to receive a daily supplement of salmon bone powder (CalGo^®^, Hofseth Biocare ASA, Ålesund, Norway) or an inert placebo (maltodextrin). Recruitment began in August 2021 and concluded in June 2023.

The trial was designed as a prevention study—accordingly, we targeted women with osteopenia—representing a population at measurable near-term risk of BMD due to age-related acceleration in bone turnover, yet not routinely eligible for pharmacologic therapy. The approach preserved ethical equipoise and allowed for sufficient sensitivity to detect clinically relevant BMD changes over 24 months, while excluding women with normal BMD (in whom change would be minimal) and those with established osteoporosis (for whom pharmacologic treatment is indicated).The protocol was reviewed and approved by the Regional Committee for Medical and Health Research Ethics (Application ID 264146), Central Norway, and all participants provided written informed consent prior to enrollment. The trial was conducted in accordance with the principles of the Declaration of Helsinki and reported following the Consolidated Standards of Reporting Trials (CONSORT) guidelines. Regular site visits by a clinical study monitor ensured adherence to protocol and data quality throughout this study.

### 2.2. Participants and Data Collection

Eligible participants were women aged ≥ 50 years with osteopenia of the femoral neck, confirmed by DXA with a T-score > −2.5 and ≤ −1.0. This range was selected to include participants with age-related bone loss suitable for preventive evaluation, consistent with the study design. Menopausal status was based on participant self-report; no formal verification was performed, and age ≥ 50 years was used as a pragmatic inclusion threshold corresponding to the typical age of menopause. Inclusion required female sex, steady body weight during the month prior to enrollment, and proficiency in Norwegian. Women with osteoporosis (T-score ≤ −2.5), a history of fragility fractures, or known fish allergy were excluded. Additional exclusion criteria were use of drugs affecting bone metabolism (e.g., glucocorticoids, hormone therapy < 6 months, long-term heparin, anticonvulsants, proton pump inhibitors, lithium, anti-osteoporotic drugs, cancer therapy, or selective estrogen receptor modulators) and medical conditions influencing bone health or nutrient absorption (e.g., Paget’s disease, osteomalacia, neoplasia, Crohn’s disease, celiac disease). After written informed consent, baseline demographic characteristics, medical history, comorbidities, and concomitant medications were recorded.

BMD was assessed at the femoral neck, lumbar spine (L1–L4), and distal radius at baseline, 12 months, and 24 months. Participants attended six on-site visits: screening, baseline (month 0), early study (3 months), 6 months, mid-study (month 12), and study completion (month 24). All scans were performed by certified densitometry technologists at participating trial centers. In Ålesund, assessments were conducted at Medi3 using a MedixDR densitometer (DMS Imaging, Gallargues-le-Montueux, France). In Kristiansund and Oslo, measurements were performed at Kristiansund Hospital and Lovisenberg Diakonale Sykehus, respectively, using locally available DXA systems in accordance with standard clinical protocols. For hip measurements, the left side was used unless prior surgery necessitated measurement of the right hip.

Safety monitoring included hematology, liver and kidney function tests, electrolytes, and serum creatinine, albumin, calcium, and vitamin D levels. Urine calcium was measured throughout follow-up. Bone turnover markers—serum N-terminal propeptide of type I collagen (P1NP) and β-C-terminal telopeptide (β-CTX-I)—were analyzed at Lab1 AS (Sandvika, Norway). At each study visit, participants reported any new or ongoing medical issues. Serious adverse events (SAEs) were defined as events that were fatal, life-threatening, required hospitalization, or resulted in significant incapacity or disability.

All SAEs were reviewed and adjudicated by the medical monitor, and fractures were prospectively designated as adverse events of special interest. Adherence was monitored at each study visit and calculated as the proportion of days supplements were taken relative to total follow-up days.

Study data were collected at trial sites and managed using Ledidi Core collaborative software (version 63d4bd8; Ledidi AS, Oslo, Norway).

### 2.3. Interventional Product and Blinding

The investigational product, CalGo^®^ (Hofseth BioCare ASA, Ålesund, Norway), is a natural marine bone powder. It is derived from the offcuts of the Norwegian aquaculture industry. Freshly filleted Atlantic salmon is placed into tanks containing water warmed to 40 °C with added natural proteolytic enzymes. The fish muscle is thereby hydrolysed, and the bones fall to the bottom of the tanks, from where they are removed, dried, and milled into a powder.

CalGo^®^ consists primarily of MCH (>19% elemental calcium, >9% elemental phosphorus) embedded within a collagen-rich matrix (>25% collagen, predominantly type II), produced without additives or harsh processing to preserve the native quality of both calcium and collagen. CalGo^®^ is thus a bone-matrix biocomposite (hydroxyapatite + collagen + native trace minerals [e.g., magnesium, phosphorus, zinc]), distinct from nano-hydroxyapatite preparations. In contrast to bovine ossein–hydroxyapatite products, CalGo^®^ is marine-sourced and manufactured without strong acid/alkali demineralization, preserving the native mineral–organic structure.

The manufacturing process follows Good Manufacturing Practices (GMP) and Hazard Analysis and Critical Control Points (HACCP) principles, with certification under FSSC 22000, and the product is Kosher and Halal compliant. CalGo^®^ has also been granted New Dietary Ingredient (NDI) status by the U.S. FDA for use in food supplements intended for human consumption.

For the trial, CalGo^®^ and placebo were encapsulated in identical opaque capsules and dispensed in indistinguishable containers to ensure blinding. Each daily dose of 2 g CalGo^®^ (administered as four capsules) provided approximately 380 mg elemental calcium, 500 mg type II collagen, and 40 µg vitamin D_3_. The comparator consisted of 2 g maltodextrin placebo, similarly, encapsulated and packaged.

### 2.4. Blood Collection and Laboratory Procedures

#### 2.4.1. Venous Draw, Sample Handling, and Processing

Fasting venous blood (~12 mL) was collected via a 21 G butterfly needle after ≥10 min of seated rest. Two tubes were drawn in standardized order: an 8 mL serum-gel tube (yellow/red top) for bone-turnover markers and clinical chemistry, and a 4 mL K_2_-EDTA tube (lavender top; VACUETTE®, Greiner Bio-One GmbH, Kremsmünster, Austria) for hematology. Tubes were gently inverted immediately after collection (5–6 times for serum, 8–10 for EDTA). The serum tube was kept upright at room temperature, protected from light, and allowed to clot for 30–60 min.

Clinical chemistry and hematology samples were processed according to hospital protocols. The serum-gel tube was centrifuged at 2000× *g* for 10 min at 4 °C, with serum stored at 4–8 °C for same-day transport to the hospital laboratory (HMR, Ålesund, Norway). The EDTA tube was not centrifuged; it was stored at room temperature if analyzed within 8 h or refrigerated at 4–8 °C if delayed, and transported together with the serum samples.

Serum for bone marker analysis was collected in a dedicated 8 mL serum-gel tube, clotted and centrifuged under the same conditions. The supernatant was pipetted into sterile 0.5 mL low-binding cryovials (minimum 2 mL per participant), snap-frozen in dry ice, and stored at −80 °C within 2 h of venipuncture, pending batch shipment to the central laboratory.

#### 2.4.2. Transport and Destination Laboratories

Serum aliquots for bone-turnover markers were shipped weekly on dry ice to Lab 1 (Medisinsk Laboratorium, Sandvika, Norway), which coordinated forwarding to SYNLAB MVZ Leinfelden, Germany. Where available, β-CTX-I was analyzed from EDTA plasma due to its higher stability; otherwise, serum was used. All bone marker samples adhered to SYNLAB’s matrix- and temperature-specific pre-analytical requirements.

Blood samples for routine clinical chemistry and hematology were delivered to Medi3 Ålesund, a local private health center, which arranged same-day courier transport to the hospital clinical chemistry department (HMR Ålesund).

#### 2.4.3. Sample Documentation and Quality Assurance

Each sample was labeled with a 2D barcode and linked to participant ID, collection time, and processing metadata. Chain-of-custody and temperature logs were maintained throughout handling, storage, and shipment. Any deviations from protocol (e.g., prolonged clotting time or delayed freezing) were documented and flagged for sensitivity analyses.

#### 2.4.4. Laboratory Assays and Reference Intervals

Bone-turnover markers were measured by electrochemiluminescence immunoassay (ECLIA) at SYNLAB Weiden. Reference intervals were 20.3–76.3 µg/L for P1NP and 0.18–1.02 µg/L for β-CTX-I, based on postmenopausal women not using hormone replacement therapy. Routine clinical chemistry and hematology were performed at HMR Ålesund using accredited automated platforms.

### 2.5. Sample Size Determination and Statistical Analysis

Sample size estimation was based on published variance data for femoral-neck BMD in postmenopausal women [18], expressed in T-score units. Assuming a conservatively rounded standard deviation of 0.50 T-score units, 45 participants per group were required to detect a 0.30-unit between-group difference with 80% power and a two-sided α = 0.05. Allowing for an anticipated 20% attrition rate, the target enrollment was 56 women per group (total *n* = 112).

Participants were randomized in a 1:1 ratio to CalGo^®^ or placebo using variable block sizes of 4 and 6, with the sequence generated by an independent statistician. Allocation was concealed from investigators, site staff, and participants; study capsules were indistinguishable in appearance and packaging.

The prespecified primary outcome was the 24-month change in areal bone mineral density (BMD; g/cm^2^) at the left femoral neck, calculated as follow-up minus baseline. Group differences were evaluated with ordinary least squares linear regression, using unadjusted and baseline-adjusted models (adjusted for baseline femoral neck BMD). Secondary skeletal outcomes—change in lumbar spine (L1–L4) BMD and change in distal radius BMD—were analyzed analogously.

Exploratory outcomes with repeated measures (e.g., bone turnover markers, liver enzymes, urinary calcium/creatinine) were assessed using linear mixed-effects models with random intercepts. Binary responder outcomes were examined using logistic regression, and adverse event frequencies were compared with χ^2^ tests.

Between-group balance in concomitant medication use at baseline was assessed descriptively (counts and proportions by group for each category). No formal statistical testing or standardized difference calculations were performed. All efficacy analyses followed the intention-to-treat principle, including all randomized participants with at least one post-baseline measurement. Statistical significance was defined as two-sided *p* ≤ 0.05. Analyses were performed in Stata/SE 17.0 (StataCorp, College Station, TX, USA).

Responder analyses were conducted post hoc to examine within-participant changes in bone mineral density (BMD). In the absence of established responder thresholds for nutritional interventions, a ≥1% increase from baseline was selected pragmatically to capture small, potentially meaningful improvements in bone density. This criterion contrasts with the 3% least significant change (LSC) commonly applied in pharmacologic trials (e.g., abaloparatide or teriparatide studies), where larger BMD gains are expected [19]. Accordingly, the 1% threshold was used as an exploratory and descriptive indicator, rather than a validated or predefined efficacy endpoint.

## 3. Results

### 3.1. Participant Characteristics and Study Completion

Eighty participants (from 346 screened) with confirmed osteopenia were enrolled across three study centers and randomized to either the CalGo^®^ group (*n* = 40) or the placebo group (*n* = 40) for a 24-month intervention. Recruitment was distributed across Ålesund (*n* = 56), Kristiansund (*n* = 11), and LDS (*n* = 13), with site-specific completion rates of 77%, 64%, and 69%, respectively. Although the original recruitment target was 112 participants, only 80 were enrolled. The shortfall was primarily due to slower-than-anticipated recruitment and a limited eligible pool, as a high proportion of screened individuals (266/346, 77%) did not meet the osteopenia inclusion criteria.

Of the 80 enrolled participants, 59 (74%) completed the full intervention, with comparable completion rates between CalGo^®^ (72.5%, *n* = 29) and placebo (75.0%, *n* = 30). Twenty-one participants (26%) discontinued prematurely, evenly distributed between the two arms: 14 within the first 6 months, 4 around the 12-month visit, and 3 after 18 months. Reasons for discontinuation included tolerability issues, initiation of osteoporosis therapy following new diagnoses or fracture, intercurrent medical conditions, and personal factors. Two participants withdrew following serious adverse events (one in each group). Participant screening, randomization, follow-up, and retention are summarized in the CONSORT flow diagram (Figure 1).

Baseline characteristics were generally well balanced between the CalGo^®^ and placebo groups, supporting effective randomization (Table 1). The overall mean age was 61.9 ± 6.9 years (range 50–80), with a near-equal distribution between participants aged 50–61 and 62–80 years. Anthropometric measures were closely aligned across groups: BMI averaged 24.8 ± 4.1 kg/m^2^ in the CalGo^®^ group and 25.0 ± 4.2 kg/m^2^ in the placebo group, values near the upper limit of the normal range.

Groups were similar in age (62.6 ± 7.1 vs. 60.5 ± 6.1 years) and femoral-neck BMD (0.74 ± 0.07 vs. 0.75 ± 0.05 g/cm^2^). P1NP (48.0 ± 17.0 vs. 54.8 ± 21.8 µg/L) and β-CTX-I (0.26 ± 0.11 vs. 0.30 ± 0.13 µg/L) were within postmenopausal reference ranges in both arms with no evident between-group differences, indicating no systematic imbalance related to menopausal status.

Bone density values were consistent with osteopenia. Femoral neck BMD and T-scores were nearly identical across groups, whereas lumbar spine BMD was modestly higher in the CalGo^®^ group (0.93 ± 0.13 vs. 0.88 ± 0.11 g/cm^2^). Bone turnover markers (P1NP and β-CTX-I) fell within expected postmenopausal ranges and showed no between-group differences.

Biochemical parameters were within reference limits and broadly comparable between groups. Serum 25-hydroxyvitamin D (vitamin D) was higher in the CalGo^®^ group than placebo (83.6 ± 30.9 vs. 73.7 ± 23.5 nmol/L; SMD = 0.36). Each group means exceeded the reference range, indicating adequate vitamin D status in the study population. EQ-VAS scores were high in both CalGo^®^ (82.15 ± 13.91) and placebo (82.90 ± 14.81), reflecting generally good self-perceived health.

Baseline concomitant medications are summarized in Table 2. Overall, the distribution of medication categories was comparable between the CalGo^®^ and placebo groups. The most frequently reported categories were allergy medications (14.3% vs. 6.7%), analgesics (12.2% vs. 2.2%), and other specified medications (53.1% vs. 57.8%) in the CalGo^®^ and placebo groups, respectively. Minor differences were observed for anticoagulants (0% vs. 6.7%) and cholesterol-lowering agents (4.1% vs. 11.1%), reflecting small absolute numbers and considered consistent with random variation. In summary, baseline concomitant medication use was well balanced across treatment groups. Adherence to the assigned intervention over 24 months was high in both groups, with mean compliance of 93.2% in the CalGo^®^ group and 88.9% in the placebo group, based on available intake records (*n* = 29 and 30, respectively).

### 3.2. Efficacy

#### 3.2.1. Primary Outcome: Change in Femoral Neck BMD (g/cm^2^)

Over 24 months, change in femoral neck BMD differed significantly (Figure 2) between treatment groups (β = 0.020 g/cm^2^, 95% CI 0.001 to 0.038; *p* = 0.038). Participants in the placebo group showed a mean decline of −0.017 g/cm^2^ (95% CI –0.030 to −0.004), whereas BMD remained stable in the CalGo^®^ group (+0.003 g/cm^2^, 95% CI −0.010 to +0.016).

Relative to baseline, femoral neck BMD increased by +0.4% in the CalGo^®^ group and declined by −2.4% in the placebo group (Table 3). The 95% confidence interval for the placebo group (−0.0297 to −0.0036 g/cm^2^) indicated consistent bone loss, whereas the interval for the CalGo^®^ group (−0.0102 to +0.0163 g/cm^2^) crossed zero and centered near stability, consistent with maintenance of femoral neck BMD over 24 months. The adjusted between-group difference was +0.0192 g/cm^2^ (95% CI 0.0006 to 0.0378; *p* = 0.044), corresponding to a relative preservation of approximately 2.8% compared with placebo.

After adjusting for baseline femoral neck BMD, the treatment effect remained significant (Table 4, β = 0.019 g/cm^2^, 95% CI 0.001 to 0.038; *p* = 0.044), while baseline values were not predictive of subsequent change (*p* = 0.319).

Relative to baseline, femoral neck BMD increased by +0.4% in the CalGo^®^ group and declined by −2.4% in the placebo group (Table 3). The 95% confidence interval for the placebo group (−0.0297 to −0.0036 g/cm^2^) indicated consistent bone loss, whereas the interval for the CalGo^®^ group (−0.0102 to +0.016).

#### 3.2.2. Secondary Outcome: Change in Lumbar Spine BMD (L1–L4)

Over 24 months, lumbar spine bone mineral density (BMD) showed divergent trajectories between groups (Table 5). Participants in the placebo group experienced a mean decline of −0.030 g/cm^2^ (95% CI −0.049 to −0.011; *p* = 0.002), whereas BMD remained stable in the CalGo^®^ group (−0.003 g/cm^2^; 95% CI −0.023 to +0.016; *p* = 0.721). The 95% confidence intervals supported these findings: the placebo group’s interval was entirely negative, consistent with bone loss, whereas the CalGo^®^ group’s interval crossed zero and centered near stability, indicating maintenance of lumbar spine BMD over 24 months.

At 12 months mean BMD was 0.92 g/cm^2^ (95% CI 0.88–0.97) in the CalGo^®^ group and 0.86 g/cm^2^ (95% CI 0.82–0.90) in the placebo group. The narrow overlap (0.88–0.90) occurred only at the margins, indicating early divergence. By 24 months, mean BMD was 0.93 g/cm^2^ (95% CI 0.88–0.98) for CalGo^®^ and 0.85 g/cm^2^ (95% CI 0.82–0.89) for placebo, with no confidence interval overlap. This lack of overlap strongly suggests a meaningful and sustained difference in lumbar spine BMD trajectories between groups.

The unadjusted between-group difference in mean change was +0.023 g/cm^2^ (95% CI −0.003 to +0.050; *p* = 0.086). After adjusting for baseline lumbar spine BMD, the treatment effect was +0.026 g/cm^2^ (95% CI −0.001 to +0.054; *p* = 0.058), corresponding to a relative preservation of approximately 3.0% compared with placebo (Table 6). Baseline values were not predictive of subsequent change (*p* = 0.313). These findings suggest attenuation of bone loss at the lumbar spine with CalGo^®^, although between-group differences did not reach statistical significance.

#### 3.2.3. Secondary Outcome: Change in Distal Radius BMD

At the distal radius, no treatment effect was observed. Over 24 months, distal radius BMD remained essentially stable in both groups. Model-adjusted mean changes were −0.0029 g/cm^2^ (95% CI −0.027 to +0.021) in the placebo group and +0.0029 g/cm^2^ (95% CI −0.022 to +0.027) in the CalGo^®^ group, consistent with the unadjusted within-group means shown in Table 7.

The between-group difference was non-significant (*p* = 0.623 unadjusted; *p* = 0.735 adjusted; Table 8). Lower baseline distal radius BMD predicted greater subsequent bone loss, independent of treatment (*p* < 0.001).

#### 3.2.4. Proportion of Responders Across Skeletal Regions

Responder analyses (≥1% BMD gain from baseline to 24 months) revealed skeletal site-specific patterns (Table 9). At the femoral neck, CalGo^®^ participants had higher odds of response than placebo (OR = 1.89, 95% CI 0.57–6.29; *p* = 0.298), though the difference was not statistically significant.

In contrast, at the lumbar spine, CalGo^®^ was associated with a significantly greater likelihood of response. Logistic regression indicated 3.6-fold higher odds of achieving ≥1% gain versus placebo (OR = 3.61, 95% CI 1.04–12.6; *p* = 0.043). Baseline BMD and age were not predictive of response.

At the distal radius, no treatment effect was observed (OR = 0.98, 95% CI 0.29–3.37; *p* = 0.980), with baseline radius BMD being the strongest predictor of outcome (*p* = 0.004).

#### 3.2.5. Secondary Outcome: Health-Related Quality of Life—EQ-5D-3L and EQ-VAS

No significant differences were observed between groups in changes from baseline in either the EQ-5D-3L index (Δ = −0.0054; *p* = 0.746) or EQ-VAS scores (*p* = 0.763). Adjustment for baseline values did not materially alter the results. Baseline EQ-5D-3L was a strong independent predictor of subsequent change (β = −0.51, *p* < 0.001), with higher initial health-related quality of life associated with smaller gains, reflecting a ceiling effect in this relatively healthy cohort.

#### 3.2.6. Serum Bone Turnover Markers (P1NP & β-CTX-I)

##### P1NP

Between-group differences in P1NP were not statistically significant (Figure 3). CalGo^®^ was associated with a mean difference of −2.98 µg/L relative to placebo (*p* = 0.59), which increased to −6.55 µg/L after adjustment for baseline (*p* = 0.18). Baseline P1NP strongly predicted subsequent change (β = −0.58, *p* < 0.001), with higher initial levels associated with greater declines. In mixed-effects models, neither treatment nor treatment-by-time interactions were statistically significant. Both groups exhibited early reductions at 2 months (placebo: 57.7 → 53.0 µg/L; CalGo^®^: 53.0 → 42.5 µg/L), a partial rebound at 3 months, and subsequent increases by 5 months. Across visits, mean P1NP levels in the CalGo^®^ group remained consistently below those in the placebo group, but differences were not statistically significant.

##### β-CTX-I

No significant treatment effect was observed for β-CTX-I. In the unadjusted model, CalGo^®^ was associated with a mean change of −0.019 µg/L relative to placebo (*p* = 0.67). After adjustment for baseline β-CTX-I, the estimated effect was −0.027 µg/L (*p* = 0.53). Baseline β-CTX-I showed a trend toward predicting subsequent change (β = −0.34, *p* = 0.07), indicating that higher initial values were associated with greater reductions. Mixed-effects regression likewise indicated no significant main or interaction effects of treatment. Over time, β-CTX-I levels remained stable at 2 months, increased at 3 months, and remained elevated at 5 months (Figure 4). At all visits, mean β-CTX-I levels in the CalGo^®^ group were numerically lower than those in the placebo group, but none of these differences reached statistical significance.

When examining coupling of bone turnover markers, positive associations were observed between change in β-CTX-I and change in alkaline phosphatase (ALP; CalGo^®^: *r* = 0.43, *p* = 0.025; placebo: *r* = 0.54, *p* = 0.003). Regression analyses indicated a steeper slope in CalGo^®^ (β = 55.6, 95% CI: 7.6–103.7, *p* = 0.025) compared with placebo (β = 30.8, 95% CI: 11.6–50.0, *p* = 0.003), suggesting a stronger ALP response relative to resorption. Baseline ALP was not associated with subsequent β-CTX-I change (*p* = 0.688).

##### Exploratory Analyses of Bone Turnover Markers and BMD Relationships

At 3 months, between-group differences in β-CTX-I change from baseline were non-significant (β = −0.032, *p* = 0.26). Baseline β-CTX-I significantly predicted change (β =−0.27, *p* = 0.024), with greater early reductions among participants with higher initial values. Marginal means indicated a slight increase in placebo (+0.016 µg/L) versus a small decrease with CalGo^®^ (−0.016 µg/L), favoring attenuated resorption in the experimental group.

At 24 months, changes in turnover markers were not associated with lumbar spine BMD in the CalGo^®^ group (P1NP: β = −5.0 × 10^−5^, *p* = 0.94; β-CTX-I: β = −0.081, *p* = 0.22). In contrast, placebo showed a significant inverse relationship between P1NP change and BMD change (β = −0.00099, *p* = 0.013), with greater P1NP increases linked to greater bone loss. β-CTX-I was not significant (β = −0.061, *p* = 0.32). These findings suggest uncoupled remodeling in placebo, versus metabolic stabilization with CalGo^®^.

Baseline ALP did not predict early changes in β-CTX-I (β = −0.00025, *p* = 0.69), indicating no evidence that formation status influenced resorption dynamics.

#### 3.2.7. Safety

##### Adverse Events

The overall incidence of adverse events across two years was low in both groups. At 12 months, 20.0% of participants in the placebo arm reported at least one event compared with 12.8% in the CalGo^®^ arm (χ^2^ = 0.70, *p* = 0.403). By study end, adverse events had declined further, occurring in 10.0% of placebo participants and 3.5% of those receiving CalGo^®^ (χ^2^ = 1.00, *p* = 0.317). Although these differences were not statistically significant, the numerical pattern favored CalGo^®^.

Across the trial, 32 adverse events were reported, occurring in 65.6% of participants (Table 10). The majority were of mild or moderate intensity, and most were judged by investigators as unrelated or unlikely to be related to study product. The most frequent categories were gastrointestinal complaints—dyspepsia, constipation, diarrhea, nausea, and bloating—while isolated cases included headache, cardiac events (palpitations/arrhythmia), dermatologic symptoms (eczema, pruritus, vesicular rash), and nonspecific complaints. A total of eight fractures were recorded (four rib, three upper extremity, one toe).

Three serious adverse events required hospitalization: paroxysmal atrial fibrillation (placebo, severe, unlikely related), ischemic cerebrovascular event (CalGo^®^, mild, unlikely related), and bacterial skin infection (placebo, moderate, not related). Two of these events (atrial fibrillation and cerebrovascular event) led to study withdrawal, whereas the infection resolved and the participant continued treatment. No deaths occurred, and no hepatic or renal safety concerns were observed during follow-up.

A total of eight fracture-related events (Table 11) were reported during the 24-month study (CalGo^®^, n = 5; placebo, n = 3). All were classified as mild to moderate in intensity and resolved without sequelae. Three cases (CalGo^®^, n = 2; placebo, n = 1) were self-reported and not clinically verified. Investigators judged all events as either “not related” or “unlikely related” to study product administration. No pattern in fracture type, location, or timing was observed.

#### 3.2.8. Liver and Renal Function Parameters

Creatinine remained stable and within the normal reference range throughout follow-up. At study end, mean creatinine was 69.1 ± 10.4 µmol/L in the CalGo^®^ group and 66.8 ± 8.9 µmol/L in the placebo group, with negligible mean changes from baseline (−0.07 ± 6.2 vs. +2.3 ± 5.4 µmol/L, respectively). Estimated glomerular filtration rate (eGFR) declined modestly in both groups (−2.0 ± 6.6 vs. −4.5 ± 7.0 mL/min/1.73 m^2^) but remained within the normal range, with no significant between-group differences.

The urinary calcium/creatinine ratio was stable across the intervention. At 24 months, mean values were 8.37 in the placebo group and 7.71 in the CalGo^®^ group (*p* = 0.63). Adjustment for baseline yielded similar results (*p* = 0.55). Mixed-effects regression across Baseline, 12 months and 24 months confirmed no group effect (*p* = 0.54), no time effect (*p* = 0.89), and no group × time interaction (*p* = 0.32). Predicted means (7.5–8.5) remained consistently within normal physiological limits, indicating no impact of CalGo^®^ on urinary calcium handling or renal mineral excretion.

Baseline alanine aminotransferase (ALT) concentrations were comparable between groups. During follow-up, values fluctuated slightly (≈22–26 U/L) but showed no significant main effect of group (*p* = 0.61), time (all *p* > 0.20), or group × time interaction (all *p* > 0.26). Estimated marginal means followed parallel trajectories, with no clinically meaningful changes. Thus, CalGo^®^ supplementation did not adversely affect liver function over 24 months.

#### 3.2.9. Subgroup Analyses on Dosing Behavior (Exploratory)

##### Divided Doses vs. Once-Daily Administration

No difference in femoral neck BMD response was observed between participants who took CalGo^®^ in divided doses and those who took it once daily. In the interaction model, dosing patterns showed no modifying effect (*p* = 0.95), indicating that efficacy was consistent regardless of dosing schedule.

##### With Food vs. Without Food

Within the overall model (*n* = 58), CalGo^®^ supplementation was associated with greater preservation of femoral neck BMD compared with placebo (β = 0.020 g/cm^2^, *p* = 0.044). Capsule intake with food showed no independent effect (*p* = 0.99). Analyses restricted to CalGo^®^ recipients likewise demonstrated no effect of food intake on treatment response (*p* = 0.68), and formal interaction testing confirmed the absence of effect modification (*p* = 0.48). These findings indicate that CalGo^®^ efficacy was unaffected by meal timing.

## 4. Discussion

This 24-month randomized controlled trial demonstrates that daily supplementation with CalGo^®^, a marine-derived salmon bone complex, preserved femoral neck BMD and attenuated lumbar spine loss in postmenopausal women with osteopenia. These results demonstrate measurable preservation of femoral-neck bone mass—a site of early postmenopausal vulnerability and high clinical relevance. Given the prevalence and fracture risk of osteopenia, targeted nutritional interventions such as CalGo^®^ may offer a meaningful preventive opportunity [20].

Placebo recipients experienced the age-expected 2–3% decline in femoral neck and lumbar spine BMD over two years, whereas CalGo^®^ significantly preserved femoral neck values and attenuated lumbar spine decline. The adjusted femoral-neck difference represented nearly a relative 3% preservation (*p* = 0.044), a clinically meaningful magnitude given that small BMD gains reduce fracture risk. Lumbar-spine effects trended similarly (*p* = 0.058). However, responder analysis indicated that nearly one-third of CalGo^®^ recipients achieved measurable improvements compared with only one in ten placebo participants, suggesting clinically meaningful benefits in a subset of participants. Health-related quality of life did not change between groups (EQ-5D-3L and EQ-VAS), and baseline EQ-5D-3L strongly predicted smaller subsequent gains, consistent with a ceiling effect in this relatively healthy cohort. These findings, derived from a rigorously conducted, double-blind, placebo-controlled study with validated densitometry and high adherence, provide a reliable basis for interpretation.

Effects differed by skeletal site, with significant benefits observed at the femoral neck and lumbar spine but not at the distal radius. The femoral neck comprises both cortical and trabecular bone, each contributing to its structural integrity and metabolic responsiveness [21,22], making it a key indicator of skeletal health. Clinically, fractures at this site carry the greatest morbidity, mortality, and economic burden among osteoporotic fractures, underscoring the importance of preserving bone mass to support healthy aging and reduce healthcare costs [23,24]. The site-specific pattern observed likely reflects differences in cortical and trabecular composition and their respective remodeling dynamics, with trabecular-rich regions generally more responsive to metabolic and nutritional influences—a pattern also supported by recent meta-analytic evidence [25].

Such patterns align with clinical and experimental evidence on OHC, which have been shown to influence bone remodeling and repair, thereby supporting the biological plausibility of interventions derived from native bone matrix [12,26]. At the cortical-dominant distal radius, participants with lower baseline BMD experienced greater 24-month loss regardless of treatment (*p* < 0.001), likely reflecting the slower remodeling kinetics and mechanical and structural stability rather than metabolic dependence of cortical bone.

Recent meta-analyses likewise concluded that calcium supplementation exerts only modest, context-dependent effects on bone mineral density—providing little benefit in healthy premenopausal women and no established role in well-nourished postmenopausal women—underscoring the importance of targeting nutritionally vulnerable populations [27,28]. A recent review similarly found that routine calcium supplementation yields only small, non-cumulative gains in BMD without reducing fracture incidence in community-dwelling adults. Given these findings and the associated safety concerns, dietary calcium intake remains the preferred approach [4]. Bone-matrix complexes may offer additional benefits by coupling mineral–collagen structure with bioactive factors (e.g., osteocalcin, IGFs, TGF-β) that support osteoblast activity [12]. Although not assayed here, CalGo^®^’s collagen-rich marine matrix likely contributes such anabolic cues, supporting the observed BMD preservation. Taken together, these observations suggest that CalGo^®^ and related bone-matrix complexes may help preserve bone mass through an integrated, physiologic delivery of minerals and matrix-derived signals, rather than through calcium mineral content alone.

Recent diet-based studies support this physiologic model of bone remodeling, showing that sustained, low-intensity nutrient delivery can stabilize turnover through homeostatic pathways. In a randomized trial, meals fortified with eggshell calcium and vitamin D—providing only moderate additional intake—modestly slowed femoral-neck bone loss without altering biochemical markers [8]. These findings reinforce the concept that steady, physiologic nutrient modulation, rather than high-dose supplementation, can help maintain remodeling equilibrium without provoking short-term biochemical shifts.

No significant between-group differences emerged for bone turnover markers (PINP, β-CTX-I). This was not unexpected, as these markers tend to be most responsive to short-term changes and generally require larger within-person shifts to exceed biological and analytical variability, whereas DXA-derived BMD reflects cumulative remodeling balance over years [29]. The absence of bone-turnover marker changes does not detract from the BMD findings, which capture longer-term remodeling balance. Exploratory signals in the CalGo^®^ group—specifically a steeper β-CTX-I–ALP relationship—suggest that higher resorption was accompanied by proportionally greater formation, consistent with enhanced coupling of remodeling processes. While hypothesis-generating, this observation raises the possibility that CalGo^®^ may influence coupling between resorption and formation rather than acting through a single pathway In placebo, increases in P1NP were inversely associated with 24-month lumbar spine BMD change, indicating that P1NP behaved like a remodeling stress marker—where higher apparent formation activity accompanied greater net bone loss—whereas no such associations were observed with CalGo^®^. These preliminary findings should be interpreted cautiously and require validation in larger studies with mechanistic endpoints.

The safety profile was favorable, with no hepatic or renal adverse effects and a low incidence of generally mild side effects. Renal and hepatic safety were reassuring: serum creatinine and eGFR remained within reference ranges in both groups, showing modest, age-expected declines over 24 months. ALT values exhibited only minor fluctuations with no significant between-group differences. Urinary calcium-to-creatinine ratios were stable throughout follow-up, indicating preserved renal calcium handling.

Over the two years, adverse events were generally mild/moderate with three serious events (two placebo, one CalGo^®^) and no deaths; two SAEs prompted withdrawal. Importantly, efficacy was unaffected by dosing behavior: neither split versus once-daily intake (*p* = 0.95) nor capsule consumption with versus without food (*p* = 0.99) modified femoral neck BMD response, with interaction testing confirming consistent effects across regimens (all *p* > 0.4). Adherence was high and comparable—93.2% in the CalGo^®^ group and 88.9% in the placebo group among participants with intake records—making it unlikely that the observed BMD effects are explained by differential compliance. Consistent with this, the dosing pattern (split vs. once-daily) and intake with or without food did not modify treatment response.

Over 24 months, eight fractures occurred (CalGo^®^, *n* = 5; placebo, *n* = 3), all mild to moderate and deemed unrelated or unlikely related to study product use. Three events (CalGo^®^, *n* = 2; placebo, *n* = 1) were self-reported, while the remainder were clinically or radiographically confirmed. All resolved without sequelae. Two CalGo^®^ and one placebo participant were withdrawn after initiating anti-osteoporotic therapy; all others completed follow-up.

Although more fractures occurred in the CalGo^®^ group, events were isolated, minor, and consistent with incidental trauma expected in an osteopenic cohort over two years. No temporal or anatomical clustering was observed, and no treatment-related pattern or safety signal emerged. Concomitant medication use was comparable between groups, primarily low-dose calcium and vitamin D, and remained stable throughout this study, suggesting no influence on fracture incidence or outcomes.

This study was not designed to evaluate fractures directly; nevertheless, the observed preservation of femoral neck BMD is clinically meaningful given the well-established relationship between bone density and fracture risk. The between-group difference of 0.019 g/cm^2^ corresponds to roughly 0.33 SD. Large epidemiological studies indicate that each 1-SD reduction in femoral neck BMD is associated with approximately a 1.5-fold higher risk of major osteoporotic fractures and a 2.0–2.6-fold higher risk of hip fractures [30]. Extrapolating from these relationships, the preservation of bone mass observed here could translate into a modest reduction in fracture risk if sustained over time. While such projections are model-based, and limited by the absence of microarchitectural data, they underscore the potential clinical relevance of maintaining BMD at fracture-prone skeletal sites.

The trial had limitations. Recruitment did not reach the original target, and it was neither designed nor powered to assess fractures, which remain a very important clinical endpoint in osteoporosis research. Participants were relatively healthy postmenopausal women with osteopenia from three Norwegian regions—enhancing internal validity but potentially limiting broader generalizability. A baseline imbalance in vitamin D, with higher levels in the CalGo^®^ group (*p* < 0.05), was observed. Both groups were vitamin D–replete, but this imbalance may still represent a potential confounding factor. Finally, although menopausal status was not formally verified, participants were verbally queried regarding their status, and a minimum age of 50 years was applied as a proxy indicator in the selection criteria. Most women enter menopause by this age [31]; however, a formal assessment would have provided greater precision and minimized potential confounding. Key strengths of the trial included its randomized, double-blind design, stringent eligibility criteria, prolonged follow-up period, independent data monitoring, and excellent participant adherence.

In summary, two years of CalGo^®^ supplementation preserved femoral neck BMD and attenuated lumbar spine loss in postmenopausal women with osteopenia. Along with its safety and biologic plausibility, these findings position CalGo^®^ as a promising marine-derived strategy for preserving bone at fracture-prone sites. They extend evidence for bone-matrix calcium preparations and support marine complexes as feasible, well-tolerated options for skeletal maintenance in nutritionally vulnerable women. Larger trials with fracture outcomes and mechanistic endpoints are warranted.

## 5. Conclusions

Daily supplementation with CalGo^®^ for 24 months preserved femoral neck BMD and attenuated lumbar spine loss in postmenopausal women with osteopenia, with a favorable safety profile. These findings support CalGo^®^ as a promising marine-derived strategy for skeletal health.

## Figures and Tables

**Figure 1 biomedicines-13-02616-f001:**
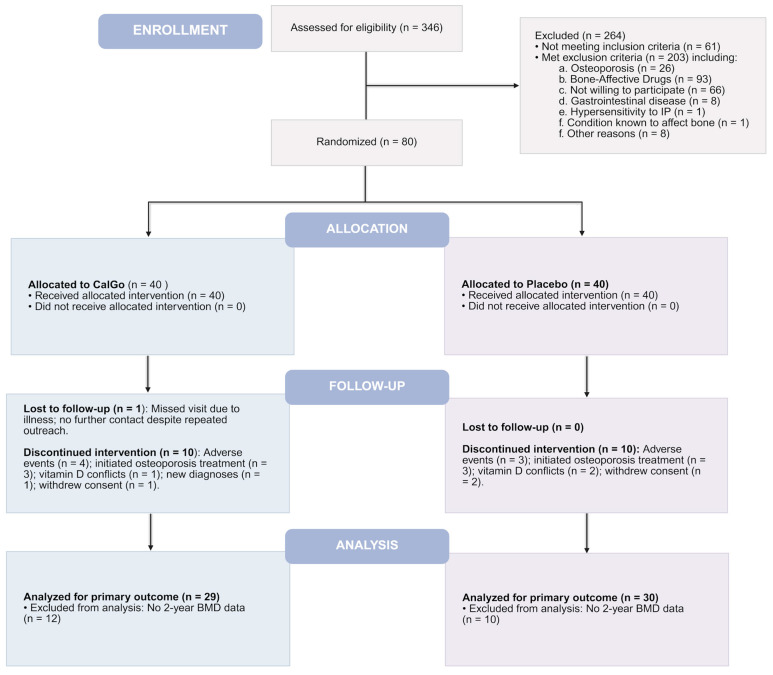
Consolidated Standards of Reporting Trials (CONSORT) flow diagram of the 24-month CalGo^®^ trial. Of 346 screened, 80 were randomized 1:1 to CalGo^®^ (*n* = 40) or placebo (*n* = 40). The diagram shows allocation, follow-up, analysis, and reasons for exclusion or discontinuation.

**Figure 2 biomedicines-13-02616-f002:**
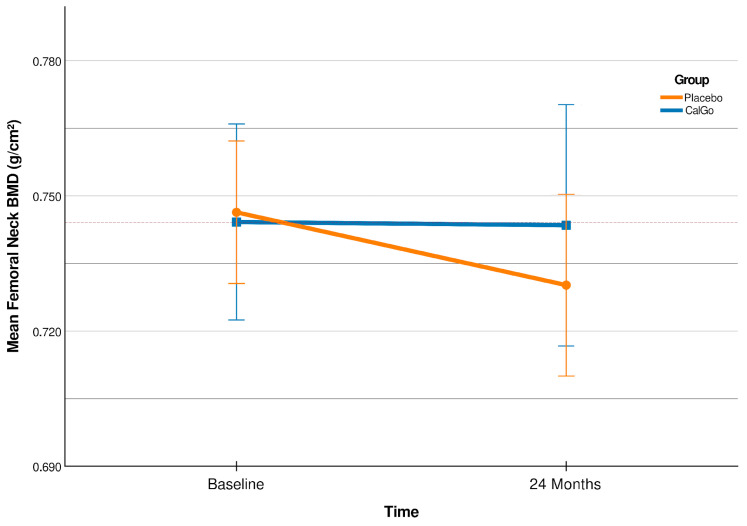
Femoral neck bone mineral density (BMD; g/cm^2^) at baseline and 24 months (CalGo^®^ *n* = 29; placebo *n* = 30 at 24 months). Means ± SD are shown; the red stippled line marks baseline CalGo^®^. The covariate-adjusted treatment effect at 24 months was +0.019 g/cm^2^ (95% CI 0.001–0.038; *p* = 0.044).

**Figure 3 biomedicines-13-02616-f003:**
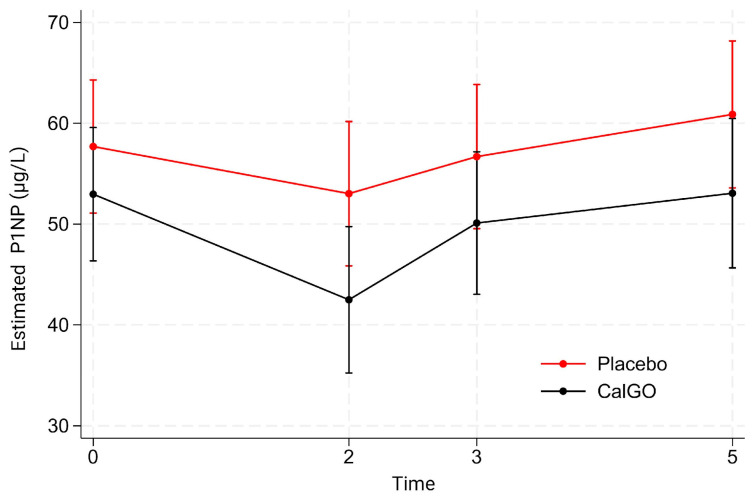
Estimated serum N-terminal propeptide of type I collagen (P1NP) concentrations (µg/L) by treatment group over time. Values are derived from a mixed-effects model with fixed effects of treatment, time, and their interaction, and random subject intercepts. Means (95% CI) are shown for placebo (red) and CalGo^®^ (black) at baseline, 2, 3, and 5 months.

**Figure 4 biomedicines-13-02616-f004:**
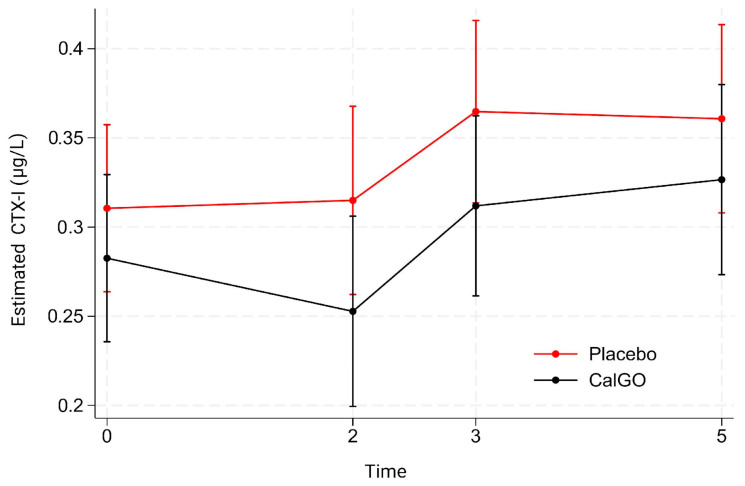
Estimated serum β-C-terminal telopeptide of type I collagen (β-CTX-I) concentrations (µg/L) by treatment group over time. Values are derived from a mixed-effects model with fixed effects of treatment, time, and their interaction, and random subject intercepts. Means (95% CI) are shown for placebo (red) and CalGo^®^ (black) at baseline, 2, 3, and 5 months.

**Table 1 biomedicines-13-02616-t001:** Baseline Characteristics by Treatment Group (Completers).

Measure	Overall (*n* = 59)	CalGo^®^ (*n* = 29)	Placebo (*n* = 30)
Demographics			
Age (years)	61.54 (±6.72)	62.62 (±7.06)	60.50 (±6.13)
Weight (kg)	68.92 (±10.68)	69.10 (±10.91)	68.73 (±10.64)
Height (m)	1.66 (±0.054)	1.66 (±0.063)	1.67 (±0.044)
Body-mass index (kg/m^2^) ^§^	24.93 (±3.81)	25.08 kg m^−2^ (±3.88)	24.79 kg m^−2^ (±3.80)
Vital Signs ^§^			
Heart Rate (bpm)	70.36 (±10.09)	70.41 (±11.97)	70.30 (±8.08)
Systolic BP (mmHg)	131.46 (±20.89)	130.79 (±16.79)	132.10 (±24.49)
Diastolic BP (mmHg)	80.56 (±11.41)	79.79 (±10.34)	81.30 (±12.49)
Bone Mineral Density			
Femoral Neck BMD (g/cm^2^)	0.74 (±0.056)	0.74 (±0.065)	0.75 (±0.047)
Femoral Neck T-score (SD)	−1.70 (±0.42)	−1.73 (±0.48)	−1.67 (±0.37)
Lumbar Spine BMD (g/cm^2^)	0.91 (±0.11)	0.93 (±0.12)	0.88 (±0.099)
Lumbar Spine T-score (SD) *	−1.55 (±0.97)	−1.32 (±1.05)	−1.77 (±0.85)
Bone Turnover Markers			
P1NP (µg/L)	51.46 (±19.69)	48.02 (±16.96)	54.78 (±21.78)
β-CTX-I (µg/L)	0.28 (±0.12)	0.26 (±0.11)	0.30 (±0.13)
Biochemistry ^†^			
Serum Calcium (mmol/L)	2.38 (±0.090)	2.40 (±0.10)	2.36 (±0.072)
Serum Phosphate (mmol/L)	1.16 (±0.18)	1.10 (±0.15)	1.22 (±0.18)
25(OH)D (nmol/L)	78.58 (±27.05)	83.62 (±30.93)	73.70 (±23.53)

Values are mean ± SD. ^†^ Reference ranges from Helse Møre og Romsdal HF (HMR), Klinikk for Diagnostikk, Ålesund. P1NP and β-CTX-I reference intervals from SYNLAB (electrochemiluminescence). § BMI, heart rate, and blood pressure ranges follow WHO and AHA guidelines. * Lumbar spine BMD and T-score (L1–L4) measured by DXA. Reference ranges: BMI 18.5–24.9; HR 60–90; SBP 90–120; DBP 60–80; Femoral neck T-score ≥ −1.0 (normal), −1.0 to −2.5 (osteopenia), ≤−2.5 (osteoporosis); Lumbar spine T-score ≥ −1.0 (normal), −1.0 to −2.5 (osteopenia), ≤−2.5 (osteoporosis); P1NP (postmenopausal, HRT 14.3–58.9; no HRT 20.3–76.3 µg/L); β-CTX-I (postmenopausal 0.18–1.02); Serum Ca 2.15–2.51; Phosphate 0.75–1.65; 25(OH)D 39–50.

**Table 2 biomedicines-13-02616-t002:** Baseline Concomitant Medications by Group (Completers).

Medication Category	Overall (*n* = 59)	CalGo (*n* = 29)	Placebo (*n* = 30)
Allergy medication	10 (10.6%)	7 (14.3%)	3 (6.7%)
Analgesics	7 (7.4%)	6 (12.2%)	1 (2.2%)
Anticoagulants	3 (3.2%)	—	3 (6.7%)
Antidepressants	—	—	—
Asthma/COPD medication	2 (2.1%)	1 (2.0%)	1 (2.2%)
Blood pressure medication	6 (6.4%)	3 (6.1%)	3 (6.7%)
Cholesterol-lowering medication	7 (7.4%)	2 (4.1%)	5 (11.1%)
Contraceptives	2 (2.1%)	1 (2.0%)	1 (2.2%)
Reflux disease/ulcer medication	2 (2.1%)	2 (4.1%)	—
Sleep medications	1 (1.1%)	—	1 (2.2%)
Other (specified in open-ended field)	52 (55.3%)	26 (53.1%)	26 (57.8%)
Missing	3 (2.3%)	1 (2.0%)	1 (2.2%)

Values are *n* (%). Categories reflect participant self-report at baseline. No formal statistical testing or standardized differences were performed.

**Table 3 biomedicines-13-02616-t003:** Femoral Neck BMD Over 24 Months.

Group	Baseline BMD (g/cm^2^)	12-Month BMD (g/cm^2^)	24-Month BMD (g/cm^2^)	Mean Change (g/cm^2^, 95% CI)	% Change
CalGo^®^	0.74 ± 0.068 g/cm^2^	0.75 ± 0.073 g/cm^2^	0.74 ± 0.070 g/cm^2^	+0.0030 (–0.0102 to +0.0163)	+0.4%
Placebo	0.75 ± 0.049 g/cm^2^	0.73 ± 0.054 g/cm^2^	0.73 ± 0.054 g/cm^2^	–0.0167 (–0.0297 to –0.0036)	–2.4%

Values are mean ± SD unless stated otherwise. % change is relative to baseline within group. Sample sizes at 24 months: CalGo^®^ n = 29; placebo n = 30.

**Table 4 biomedicines-13-02616-t004:** Between-Group Difference in Femoral Neck BMD at 24 Months.

Model	Difference (g/cm^2^)	95% CI	*p*-Value	Relative Effect ^2^
Unadjusted	+0.0197	0.0011 to 0.0383	0.038	-
Adjusted ^1^	+0.0192	0.0006 to 0.0378	0.044	+2.8%

^1^ Adjusted for baseline femoral neck BMD. ^2^ Relative effect = adjusted difference as % of baseline mean BMD.

**Table 5 biomedicines-13-02616-t005:** Lumbar Spine (L1–L4) BMD Over 24 Months.

Group	Baseline BMD (g/cm^2^)	12-Month BMD (g/cm^2^)	24-Month BMD (g/cm^2^)	Mean Change (g/cm^2^, 95% CI)	% Change
CalGo^®^	0.93 ± 0.13	0.92 ± 0.13	0.93 ± 0.13	–0.0051 (–0.0240 to +0.0138)	–0.4%
Placebo	0.88 ± 0.11	0.86 ± 0.10	0.85 ± 0.096	–0.0282 (–0.0468 to –0.0096)	–3.4%

Values are mean ± SD unless stated otherwise. % change is relative to baseline within group. Region: L1–L4 by DXA.

**Table 6 biomedicines-13-02616-t006:** Between-Group Difference in Lumbar Spine BMD at 24 Months.

Model	Difference (g/cm^2^)	95% CI	*p*-Value	Relative Effect ^2^
Unadjusted	+0.023	−0.003 to +0.050	0.086	–
Adjusted ^1^	+0.026	−0.001 to +0.054	0.058	+3.0%

^1^ Adjusted for baseline femoral neck BMD. ^2^ Relative effect = adjusted difference as % of baseline mean BMD.

**Table 7 biomedicines-13-02616-t007:** Distal Radius BMD Over 24 Months.

Group	Baseline BMD (g/cm^2^)	12-Month BMD (g/cm^2^)	24-Month BMD (g/cm^2^)	Mean Change (g/cm^2^, 95% CI)	% Change
CalGo^®^	0.66 ± 0.13	0.66 ± 0.12	0.66 ± 0.13	+0.0049 (−0.0233 to +0.0332)	+0.4%
Placebo	0.65 ± 0.11	0.67 ± 0.090	0.66 ± 0.088	−0.0049 (−0.0326 to +0.0229)	−0.4%

Values are mean ± SD unless stated otherwise. % change is relative to baseline within group.

**Table 8 biomedicines-13-02616-t008:** Between-Group Difference in Distal Radius BMD at 24 Months.

Model	Difference (g/cm^2^)	95% CI	*p*-Value	Relative Effect ^2^
Unadjusted	+0.0098	−0.030 to +0.049	0.623	–
Adjusted ^1^	+0.0058	−0.028 to +0.040	0.735	+0.9%

^1^ Adjusted for baseline femoral neck BMD. ^2^ Relative effect = adjusted difference as % of baseline mean BMD.

**Table 9 biomedicines-13-02616-t009:** Responder Analysis: ≥1% BMD Gain at 24 Months.

Site	Odds Ratio (95% CI)	*p*-Value (OR)
Femoral Neck	1.89 (0.57–6.29)	0.298
Lumbar Spine	3.61 (1.04–12.6)	0.043
Distal Radius	0.98 (0.29–3.37)	0.980

Responder defined a priori for this exploratory analysis as ≥1% increase from baseline. Odds ratios from logistic regression adjusted for baseline BMD and age.

**Table 10 biomedicines-13-02616-t010:** Adverse Events Through 24 Months.

Category	Placebo (*n* = 20)	CalGo^®^ (*n* = 12)	Total (*n* = 32)
Any adverse event (AE)	15 (75.0%)	6 (50.0%)	21 (65.6%)
Fractures	3 (15.0%)	5 (41.7%)	8 (25.0%)
Serious adverse events (SAE)	2 (10.0%)	1 (8.3%)	3 (9.4%)
Most common AEs			
– Dyspepsia	2 (9.1%)	1 (8.3%)	3 (8.8%)
– Constipation	1 (4.5%)	1 (8.3%)	2 (5.9%)
– Diarrhea	2 (9.1%)	0	2 (5.9%)
– Nausea	2 (9.1%)	0	2 (5.9%)
– Any cardiac event	2 (9.1%)	0	2 (5.9%)
Causality (AE, investigator-judged)			
– Possible/likely	4 (20.0%)	5 (41.7%)	–
– Unlikely/not related	14 (70.0%)	6 (50.0%)	–
Intensity of AE			
– Mild	10 (50.0%)	7 (58.3%)	–
– Moderate	8 (40.0%)	4 (33.3%)	–
Outcome of AE			
– Resolved	17 (85.0%)	11 (91.7%)	–

Values are n (%). SAE is defined as fatal/life-threatening, requiring hospitalization, or causing significant incapacity/disability. Causality assessed by investigators (possible/likely vs. unlikely/not related). Intensities classified as mild/moderate/severe.

**Table 11 biomedicines-13-02616-t011:** Fracture Events Through 24 Months.

Group	Fracture Type/Location	Intensity	Causality	Outcome	Notes
CalGo^®^ (*n* = 5)	Possible fracture of the second toe (second digit, right foot)	Mild	Not related	Resolved	Self-reported; no clinical verification.
	Right anterior rib fracture	Moderate	Unlikely	Resolved	Clinically verified.
	Proximal humeral (shoulder) fracture	Moderate	Unlikely	Resolved	Clinically verified.
	«Right rib fractures»	Moderate	Not related	Resolved	Self-reported; no clinical verification.
	Left distal radius (wrist) fracture	Moderate	Not related	Resolved	Clinically verified.
Placebo (*n* = 3)	Left rib fracture	Moderate	Unlikely	Resolved	Clinically verified.
	Left humeral (upper arm) fracture	Moderate	Unlikely	Resolved	Clinically verified.
	Right rib fracture	Moderate	Unlikely	Resolved	Self-diagnosed rib fracture; not verified clinically.

Events include rib, upper extremity, and toe fractures. Verification status (clinical/radiographic vs. self-reported) and causality per investigator judgment.

## Data Availability

The data generated and analyzed in this trial are stored securely in accordance with the General Data Protection Regulation (GDPR) and the requirements of REK Central. Due to ethical and legal restrictions, the full dataset cannot be made publicly available. However, certain de-identified individual participant data, together with a data dictionary, may be made available upon reasonable request to the corresponding author (chbj@hofsethbiocare.no), subject to GDPR and REK midt approval and a data use agreement. The statistical analysis code (Stata do-files) can likewise be provided upon request. The full trial protocol is accessible at ClinicalTrials.gov (Identifier: NCT05066477).

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
