# Peer review of "Supplementation with a Salmon Bone Complex (CalGo^®^) Preserves Femoral Neck BMD and Attenuates Lumbar Spine Loss: A 24-Month Randomized, Placebo-Controlled Trial"

_biomedicines, 2025, doi:10.3390/biomedicines13112616_

Round 1
Reviewer 1 Report
Comments and Suggestions for Authors
Supplementation with CalGo® Salmon Bone Complex Improves Femoral Neck and Preserves Lumbar Spine Bone Mineral Density: 2-Year Randomized Placebo-Controlled Trial
The study aims to evaluate the effects and safety of salmon bone powder (SBP; CalGo®, Hofseth BioCare ASA) on bone health in osteopenic women. Overall, the manuscript is informative and addresses an important research question. The following suggestions may help improve clarity and presentation:
- The introduction is well-referenced, but some paragraphs feel loosely connected. Improving the transitions would strengthen the overall flow and coherence.
- A brief introduction of CalGo® should be added. Is it a bio-composite of the calcium hydroxyapatite (micro or nano form), collagen, and trace minerals?? The formulation is close to Ossium, a bio-composite of ossein and hydroxyapatite??
- The adverse event of the fractures, which was reported that the treatment group with CalGo® (n=5; 41.7%) was higher than that of the placebo group (n=3; 15.0%) will be discussed.
- Standardize the use of time abbreviations (e.g., use h for hours and min for minutes).
- Recheck the consistency of abbreviations. For example, terms such as bone mineral density (BMD) should be introduced once with the full name, followed by the abbreviation throughout.
- Table titles (e.g., Table 1) could be more concise, with details such as data format, analysis approach, and reference ranges placed in the footnotes instead.
- Page 9, line 332: a parenthesis appears to be missing in for (−0.0102 to +0.016).
- Ensure that p in statistical values (e.g., p < 0.05) is consistently italicized.
Author Response
Please see attached peer-review response letter (reviewer 1) in pdf-format.

Reviewer 2 Report
Comments and Suggestions for Authors
This study was therefore undertaken to evaluate the effects and safety of a SBP (CalGo®, Hofseth BioCare ASA) on bone health in osteopenic women.
The introduction is correct, with appropriate bibliographical references. However, the authors should justify why the study was conducted on osteopenic women rather than on the normal population or osteoporotic women.
Methods. The methodology is described in detail and could be replicated by another research group. The sample size calculation is interesting.
It should be indicated whether the women included were menopausal, what criteria were used to define them, and whether there were differences between the groups.
The drugs they were taking should be indicated, as well as whether there were differences between the groups.
The results are clearly expressed and easy to follow. The authors divide patients into responders and non-responders based on whether they achieved a 1% change in bone mass, but they do not explain why they chose this cutoff point. It is not indicated whether this corresponds to the minimum significant change or is based on other criteria.
The discussion is clear and consistent with the observed results, indicating the weaknesses of the study.
Author Response
Please see attached peer-review response letter in pdf-format (reviewer 2).

Round 2
Reviewer 2 Report
Comments and Suggestions for Authors
The authors have answered by the authors